# GammaFocus: An image augmentation method to focus model attention for classification

**Ana Leni Frei**[1]                                    ANA.FREI@UNIBE.CH
**Amjad Khan**[1]                                    AMJAD.KHAN@UNIBE.CH
**Philipp Zens**[1]                                    PHILIPP.ZENS@UNIBE.CH
**Alessandro Lugli**[1]                              ALESSANDRO.LUGLI@UNIBE.CH
**Inti Zlobec**[1]                                    INTI.ZLOBEC@UNIBE.CH
**Andreas Fischer**[2,3]                              ANDREAS.FISCHER@UNIFR.CH

[1] *Institute of Tissue Medicine and Pathology, University of Bern, Switzerland*

[2] *Document, Image and Video Analysis Research Group, University of Fribourg, Switzerland*

[3] *iCoSyS, University of Applied Sciences and Arts Western Switzerland, Fribourg, Switzerland*

**Editors:** Under Review for MIDL 2023

## Abstract

In histopathology, histologic elements are not randomly located across an image but organize into structured patterns. In this regard, classification tasks or feature extraction from histology images may require context information to increase performance. In this work, we explore the importance of keeping context information for a cell classification task on Hematoxylin and Eosin (H&E) scanned whole slide images (WSI) in colorectal cancer. We show that to differentiate normal from malignant epithelial cells, the environment around the cell plays a critical role. We propose here an image augmentation based on gamma variations to guide deep learning models to focus on the object of interest while keeping context information. This augmentation method yielded more specific models and helped to increase the model performance (weighted F1 score with/without gamma augmentation respectively, PanNuke: 99.49 vs 99.37 and TCGA: 91.38 vs. 89.12, $p < 0.05$).

**Keywords:** digital pathology, gamma correction, image augmentation, contrast enhancement, image classification

## 1. Introduction

Digital Pathology whole slide images (WSI) are large images that need to be divided into smaller patches in order to apply deep learning methods for classification of histologic elements (Lee K and AC, 2021; Janowczyk and Madabhushi, 2016). When performing classification of small tissue fractions, such as cells, the optimal crop size around the region of interest (ROI) can be difficult to estimate. The object of interest can be difficult to classify by itself and might need contextual tissue information to make the proper decision. For that reason, the crop size should be optimized to find the correct balance between the amount of information coming from the object of interest and surrounding context information. In this work, we show that for cell-based classification, the context of the cells matters and use epithelial cell classification in colorectal tissue as an example. We first evaluated different patch sizes around the ROIs and showed that performance increased with larger patch sizes. However, the information originating from the object of interest

might be diluted when taking patches much larger than the object itself. To address this undesired effect, we propose here an image augmentation based on gamma variations to increase the contrast in the region of the object to be classified and decrease the brightness of its surroundings (Nateghi et al., 2021). We show that by using this gamma focusing, the classification accuracy significantly improve while guiding the model to focus on that area of interest even when using large patch sizes.

## 2. Materials and Methods

The dataset was composed of epithelial cells extracted from Lizard dataset, as well as epithelial cells annotated by experts on TCGA and our Institute's cohort (Graham et al., 2021). This resulted in 66'034 normal cells and 119'013 malignant cells for training and validation. Cells extracted from TCGA and PanNuke (subset from Lizard) were kept as unseen test data with 12'751 normal cells and 17'717 malignant cells. All the images were used at 20X (0.5 µm/pixel) and were cropped around cell centroids to extract cell patches.

ResNet18 and MobileNet were trained for normal versus malignant epithelial cell classification (He et al., 2015; Sandler et al., 2019). As the morphology of malignant cells can vary in size, the smallest patch size used was 32×32px in order to get the complete cell in the patch. The models were trained for different patch sizes around the cells: 32×32px, 64×64px and 128×128px. All models were trained using a 5 fold cross-validation. Table 1 shows the model's performance for the different patch sizes. As the highest F1 score was achieved using 128×128px patches, these results were analysed with GradCam and further improved using the GammaFocus augmentation. Finally, ViT16 was trained to compare the performance using a model made to retain spatial structural information (Dosovitskiy et al., 2021). Models were compared using the statistical McNemar test for paired samples.

**GammaFocus**: The GammaFocus (GF) augmentation rely on the gamma correction to adjust contrast in image analysis (Somasundaram and Kalavathi, 2012; Rahman et al., 2016). This correction is a non-linear transformation that encodes the brightness of the image. It is based on the following power law expression:

$$I_{out} = I_{in}^{\gamma}$$

where $\gamma$ encodes the changes in brightness. $\gamma > 1$ implies a gamma expansion and thus increases the contrast, as $\gamma < 1$ is a gamma compression and reduces the contrast. For RGB images, the gamma augmentation is applied per-channel.
In our experimental setup, the brightness in the center 64×64px of the 128×128px input patches was increased, as the bightness surrounding this central region was decreased. For that we used $\gamma = 1.5$ and $\gamma > 0.5$ respectively. GF augmentation on H&E cell patches can be seen in Figure 1.

During the training process, multiple other image augmentations were applied (rotation, flip, stain variations) before applying the GF transform.

**GradCam**: GradCam method was used to highlight regions impacting most the model's decision when trained with and without GammaFocus (Selvaraju et al., 2019; Gildenblat and contributors, 2021). GradCam heatmaps overlayed on the input images can be seen in Figure 1.

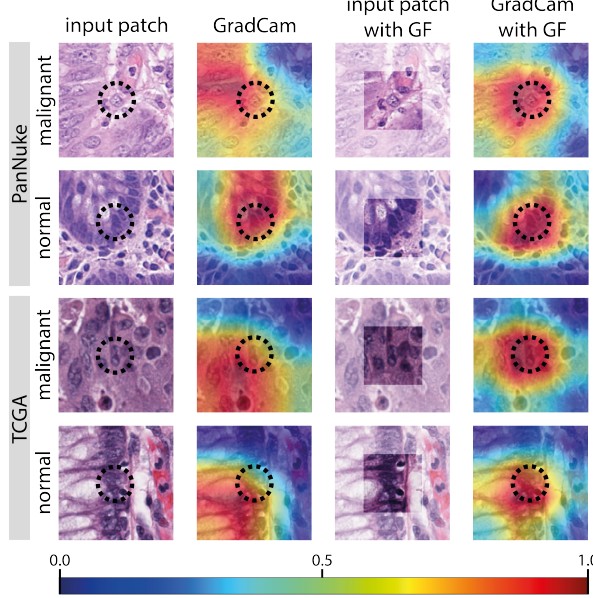

Figure 1: GradCam heatmap over cell patches from PanNuke and TCGA, with/without GF for ResNet18. Black dotted circles highlight the cell of interest.

| Patch size (px) | PanNuke | TCGA |
|---|---|---|
| 32×32 | 91.00 ± 0.006 | 68.12 ± 0.01 |
| 64×64 | 96.99 ± 0.002 | 81.43 ± 0.016 |
| **128×128** | **98.58 ± 0.04** | **89.12 ± 0.12** |

Table 1: ResNet18 weighted F1 score for different patch sizes around the epithelial cells for normal vs. malignant binary classification.

| Method | PanNuke | TCGA |
|---|---|---|
| ViT16 | 99.10 ± 0.004 | **89.47 ± 0.004** |
| ViT16 + GF | **99.31 ± 0.003** | 88.55 ± 0.01 |
| MobileNet | 99.37 ± 0.004 | 89.39 ± 0.01 |
| MobileNet + GF | **99.49 ± 0.002** | **90.13 ± 0.01** |
| ResNet18 | 98.58 ± 0.04 | 89.12 ± 0.12 |
| ResNet18 + GF | **99.13 ± 0.08** | **91.38 ± 0.17** |

Table 2: Weighted F1 score for different methods with and without GammaFocus (GF) for normal vs. malignant binary classification.

## 3. Results

Classification accuracy increased with the patch size, Table 1. However, GradCam heatmaps show that the model did not necessarily use the cell of interest for the classification decision. Upon GF augmentation, the model paid more attention to the cells of interest, see Figure 1, and the classification F1 score also increased, performing significantly better than models trained without GF, as can be seen in Table 2, $p < 0.05$ for ResNet18 and MobileNet. Best results were obtained by applying GF during training and inference.

## 4. Discussion and Conclusion

The GF augmentation method allows to take larger crops around the ROI while guiding the model to focus mainly on the object to be classified and increase models' performances. The behaviour of ViT16 with GF is not as clear as ResNet18 and Mobilenet and should be explored further in future work.

## Acknowledgments

This work was funded by the Swiss National Science Foundation (CRSII5_193832). Results presented here are based upon data generated by the TCGA Research Network.

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
