# OpenReview forum: "GammaFocus: An image augmentation method to focus model attention for classification"
_MIDL.io/2023/Short_Paper_Track — MIDL 2023 Short paper track Poster_

### Official Review · Reviewer_5QQe · 2023-04-11
**Review on "GammaFocus: An image augmentation..."**

**Rating:** 7
**Confidence:** 4

**Review:**

In this work, the authors focus on cell classification in histopathology image. Their goal is to show that the environment around the cell plays a critical role. Their use case is epithelial cell classification in colorectal tissue.

They experiment different patch sizes around the cells with various 3 classification networks.
More importantly, they propose to apply a gamma correction method (focus on a centered subpatch) on the images before data augmentation. In terms of F1 scores, improvement is marginal (however statistically significant) but the heatmaps provided by GradCAM show the model is more focused on the cell itself.

The paper is clearly written. Even though results are not groundbreaking, the GammaFocus could be interesting for future works.

However I have a question regarding the use of the GammaFocus method.

I wonder whether the GammaFocus method is applied only at training or during inference as well, as shown in Figure 1, GradCAM being applied posthoc. It would mean that it is unclear if the impact of GammaFocus comes from data augmentation during training or being applied on the test images. Maybe I misunderstood, please correct accordingly.

The reader unfamiliar with histopathological images and split into patches can wonder how you can make sure that the patchs are centered on the object of interest? since this is key to your GradCAM analysis. This question is also in relation with fig 1 where « black dotted circles highlight the cell of interest »:  there are actually many cells that can be distinguished inside the circle.

minor comments or typos
====================

- Insititute’s

- caption of figure1: « GradCam heatmap over malignant
and normal cell patches, from
PanNuke and TCGA respectively,
with/without GF for ResNet18. » It is unclear which one is which.

- p < 0.05: please specify what type of statistical test was used.

---

### Official Review · Reviewer_xYbd · 2023-04-25
**simple, promising approach; small performance gains**

**Rating:** 7
**Confidence:** 3

**Review:**

This paper focuses on cell classification from whole-slide images. The proposed method is an augmentation technique that involves increasing the contrast within in an object while decreasing the brightness of its surroundings.

-	This approach offers a simple means of focusing on the object of interest while preserving information about the surrounding context.

-	I am not certain of the novelty of this idea within this application domain, but the proposed approach seems to be reasonable and straightforward.

-	The addition of this augmentation method is found to improve classification accuracy in most cases, although the gains appear rather small (as the performance was already high without it).